# In situ X-ray-assisted electron microscopy staining for large biological samples

Sebastian Ströh[1†], Eric W Hammerschmith[1†], David W Tank[1], H Sebastian Seung[1], Adrian Andreas Wanner[1,2]*

[1]Princeton Neuroscience Institute, Princeton University, Princeton, United States;
[2]Paul Scherrer Institute, Villigen, Switzerland

**Abstract** Electron microscopy of biological tissue has recently seen an unprecedented increase in imaging throughput moving the ultrastructural analysis of large tissue blocks such as whole brains into the realm of the feasible. However, homogeneous, high-quality electron microscopy staining of large biological samples is still a major challenge. To date, assessing the staining quality in electron microscopy requires running a sample through the entire staining protocol end-to-end, which can take weeks or even months for large samples, rendering protocol optimization for such samples to be inefficient. Here, we present an in situ time-lapsed X-ray-assisted staining procedure that opens the 'black box' of electron microscopy staining and allows observation of individual staining steps in real time. Using this novel method, we measured the accumulation of heavy metals in large tissue samples immersed in different staining solutions. We show that the measured accumulation of osmium in fixed tissue obeys empirically a quadratic dependence between the incubation time and sample size. We found that potassium ferrocyanide, a classic reducing agent for osmium tetroxide, clears the tissue after osmium staining and that the tissue expands in osmium tetroxide solution, but shrinks in potassium ferrocyanide reduced osmium solution. X-ray-assisted staining gave access to the in situ staining kinetics and allowed us to develop a diffusion-reaction-advection model that accurately simulates the measured accumulation of osmium in tissue. These are first steps towards *in silico* staining experiments and simulation-guided optimization of staining protocols for large samples. Hence, X-ray-assisted staining will be a useful tool for the development of reliable staining procedures for large samples such as entire brains of mice, monkeys, or humans.

**\*For correspondence:**
adrian.wanner@psi.ch

*These authors contributed equally

## Editor's evaluation

This important study explores the kinetics of heavy metal staining of tissue using time-lapse imaging with X-ray micro computed tomography (CT). Introducing a compelling approach to investigate staining in situ, this work will be of interest to the wide community of scientists preparing biological samples in particular for large-volume electron microscopy. It will become a reference for the field in establishing a quantitative tool for assessing and developing staining protocols.

## Introduction

In the past decade the image acquisition rates of biological electron microscopy facilities have been scaled to $10^7$–$10^9$ pixels per second through parallelization and automation (*Hayworth et al., 2014*; *Schalek et al., 2011*; *Ren and Kruit, 2016*; *Denk and Horstmann, 2004*; *Hayworth et al., 2020*; *Eberle et al., 2015*; *Graham et al., 2019*; *Xu et al., 2017*). The increase in imaging throughput for biological samples has been mainly driven by the emerging Neuroscience field of connectomics which aims to densely reconstruct neuronal circuits with synaptic resolution (*Briggman et al., 2011*; *Helmstaedter et al., 2013*; *Bock et al., 2011*; *Zheng et al., 2018*; *Scheffer et al., 2020*; *Kornfeld*

*et al., 2017*; *Wanner and Friedrich, 2020*; *Lee et al., 2016*; *Kasthuri et al., 2015*; *Wilson et al., 2019*; *Schmidt et al., 2017*; *Svara et al., 2018*; *Vishwanathan et al., 2017*). Also, the analysis of the terabyte-sized electron microscopy datasets produced by these studies is becoming increasingly automated using machine-learning and computer vision (*Januszewski et al., 2018*; *Dorkenwald et al., 2020*; *Schubert et al., 2019*; *Dorkenwald et al., 2017*; *Berning et al., 2015*; *Staffler et al., 2017*; *Jain et al., 2010*; *Buhmann et al., 2020*; *Vergara et al., 2020*; *Turner et al., 2020*). Both the image acquisition as well as the image analysis for these types of datasets are being scaled by parallelization to large samples on the order of several cubic millimeters or even entire brains. However, a remaining obstacle is the lack of reliable tissue processing and staining protocols for large samples where the smallest dimension is greater than 1 mm.

Existing *en bloc* electron microscopy staining protocols have been optimized for staining small samples with dimensions of less than 1 mm (*Genoud et al., 2018*; *Hua et al., 2015*; *Deerinck et al., 2010*; *Tapia et al., 2012*). Using aldehyde-stabilized cryopreservation (*McIntyre and Fahy, 2015*) the cellular ultrastructure can be preserved even in large tissue blocks with dimensions exceeding 1 mm and entire brains. Despite pioneering work on *en bloc* staining protocols for whole mouse brains (*Mikula and Denk, 2015*; *Mikula et al., 2012*), large *en bloc* stained samples still suffer from artifacts such as inhomogeneous staining and membrane or tissue cracks.

*En bloc* sample preparation for electron microscopy generally requires tissue fixation, staining and embedding in resin. First, the macromolecules in the tissue are stabilized via crosslinking by diffusing or perfusing buffered solutions of fixatives such as formaldehyde (*Claude and Fullam, 1945*; *Fox et al., 1985*) and glutaraldehyde (*Sabatini et al., 1964*). Because biological tissue is composed mostly of carbon and other low atomic number elements, the tissue is stained with heavy metals to increase the contrast in electron micrographs (*Bahr, 1954*; *Porter et al., 1945*). These heavy metal stains are composed of electron dense atoms such as osmium, lead or uranium. In addition, some of these heavy metals (e.g. osmium) also act as fixatives (*Bahr, 1955*). Finally, the stained tissue gets dehydrated and embedded in resin. The most commonly used resins are epoxy resins (*Maaløe and Andersen, 1956*), such as araldite (*Glauert et al., 1956*) or Epon (*Finck, 1960*) because of their thermal stability and electron transparency. To inspect the staining quality in an electron microscope, ultrathin sections (<100 nm) are typically collected from the embedded tissue using an ultramicrotome.

Many steps in the classic *en bloc* electron microscopy staining protocols are based on passive diffusion of chemicals into the tissue. Passive diffusion is one of the main bottlenecks for preparing large samples (smallest dimension >1 mm) for electron microscopy (*Burkl and Schiechl, 1968*; *Medawar, 1941*). For small samples (smallest dimension <1 mm) a typical *en bloc* staining protocol takes about 10–15 days including sectioning (*Tapia et al., 2012*). However, for large samples or whole brains, a diffusion-based staining protocol takes several weeks or even months (*Mikula et al., 2012*; *Mikula and Denk, 2015*; *Masís et al., 2018*).

To date, electron microscopy staining is a 'black box' and changes to staining protocols can only be assessed using an electron microscope, which in turn requires a sample to be run through the entire staining protocol. Conventional approaches for optimizing the parameters of staining protocols rely on sequential screening of hundreds of samples that have been processed end-to-end. However, sequential screening is very inefficient for months-long protocols.

Recently, X-ray based computed micro-tomography (µCT) has been introduced for a relatively fast, macroscopic assessment of staining quality and tissue integrity of resin-embedded whole mouse brains (*Mikula and Denk, 2015*; *Kuan et al., 2020*; *Dyer et al., 2017*). (*Mikula and Denk, 2015*) showed that the pixel intensities of serial-section EM images and the intensity of the corresponding reslice are similar. Building on this pioneering work, we developed in situ time-lapsed X-ray-assisted staining to observe the staining process while the samples are in the staining solution (*Figure 1a*). We used X-ray-assisted staining to explore the micro-scale tissue mechanics and the kinetics of the heavy metal diffusion and accumulation in large aldehyde-fixed brain tissue blocks, resulting in new insights on how different staining agents affect the tissue. X-ray-assisted staining opens the 'black box' of electron microscopy staining protocols. Each staining step can be monitored and assessed in real time. This enables in silico optimization of electron microscopy staining protocols, which will be particularly useful for the development of staining procedures for large biological samples such as whole brains (*Figure 1g*).

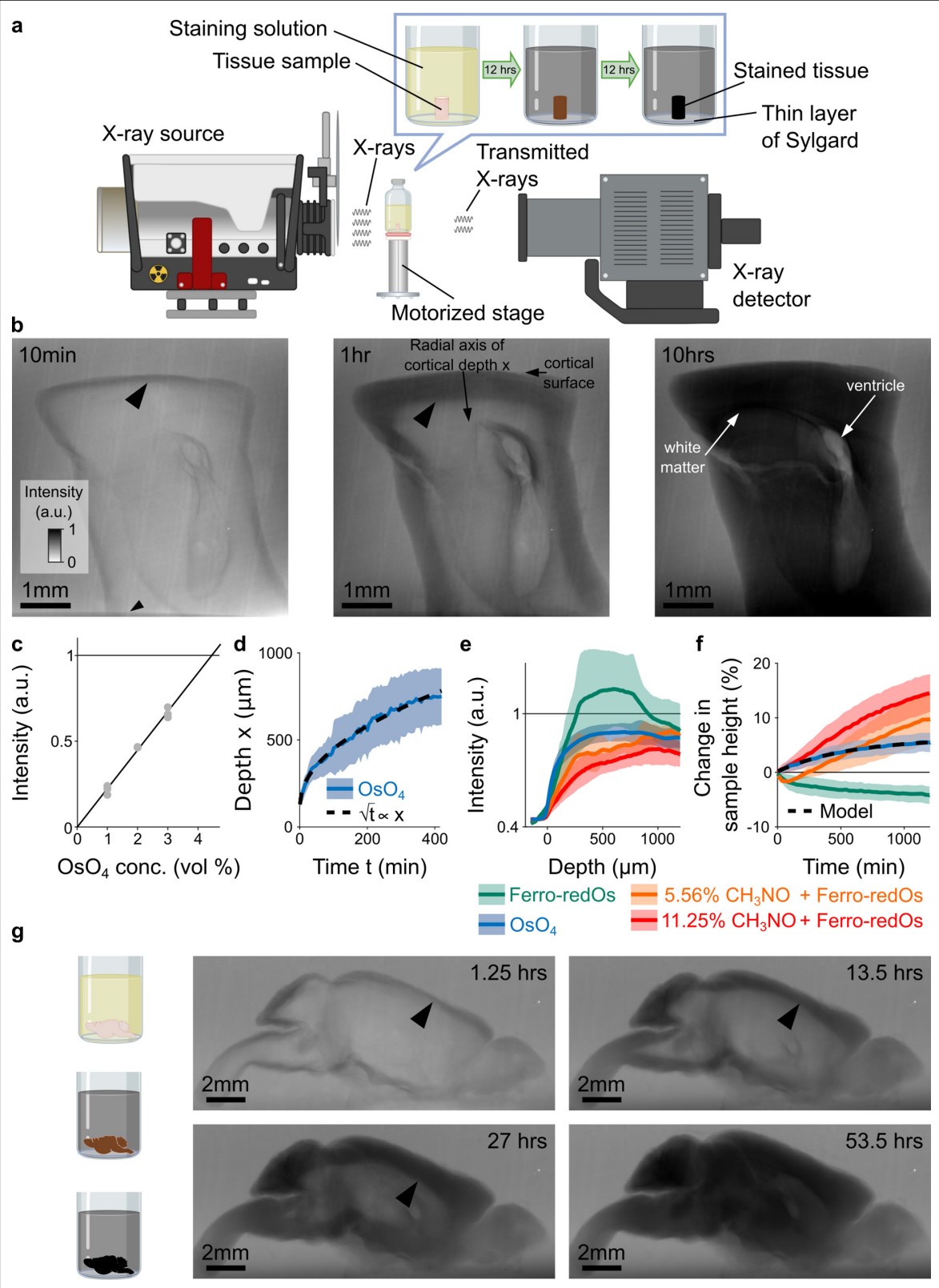

**Figure 1.** X-ray assisted staining. (**a**) Experimental setup. The sample was glued into a thin layer of Sylgard and sits in staining solution in a sealed glass vial on a motorized stage. X-rays are emitted from an X-ray source, pass through the sample, and the spatial distribution of the transmitted X-rays is detected and used to form a projection image of the X-ray absorption in the sample. Over time, the heavy metals diffuse into the tissue and the accumulation of heavy metals change the X-ray absorption properties of the sample. For the quantitative results presented in this study we used

*Figure 1 continued on next page*

*Figure 1 continued*

cylindrical tissue samples extracted with a 4 mm diameter biopsy punch from the cortex of mice. (**b**) Projection images of a 4 mm punch of mouse cortex incubated in 2% buffered osmium tetroxide ($OsO_4$) solution after 10 min, 1 hr, and 10 hr. The osmium diffuses passively from the cortical surface and the brain ventricles into the tissue and forms a staining front of tissue-bound osmium (large arrowhead) that moves towards the center of the sample as time progresses. The accumulation of heavy metals results in an increase in the pixel intensity (a.u.) of the X-ray projection image, which was measured along the radial axis of cortical depth from the cortical surface towards the white matter. The small arrowhead indicates the upper edge of the 2 mm thin layer of Sylgard. (**c**) Measured pixel intensities of the X-ray projection images for different concentrations of buffered $OsO_4$ solutions. The higher/darker the intensity, the more X-rays are absorbed. The pixel intensity/X-ray absorption scales linearly with the concentration/density of osmium (black line, Pearson correlation coefficient $r$=0.99, <$10^{-6}$; n=3 for each concentration). (**d**) Propagation of the staining front in 4 mm brain punches immersed in 2% buffered $OsO_4$ solution (n=13) measured along the radial axis of cortical depth (mean ±s.d.). The propagation can be fitted by a quadratic model (dashed black lines) in which the penetration depth x of the staining front is proportional to the square root of the incubation time t (residual standard error: $SE_{res}$ = 14.68 μm). (**e**) Intensity profiles of the spatial heavy metal accumulation after t=20 hr of incubation in 2% buffered $OsO_4$ (blue, n=13), Ferro-redOs (green, n=6). Ferro-redOs with 5.56% (orange, n=4) and 11.25% formamide (red, n=4) solutions (mean ±s.d.). In Ferro-redOs the heavy metal staining is not homogenous and there is a more densely stained tissue band at a depth of 300–800 μm. (**f**) Tissue expansion quantified as change in sample height of aldehyde-fixed brain punches immersed in $OsO_4$ (blue, n=13), Ferro-redOs (green, n=6), Ferro-redOs with 5.56% (orange, n=4) and 11.25% formamide (red, n=4), respectively, as a function of the immersion duration (mean ±s.d.). All solutions were buffered by cacodylate. A monomolecular growth model was fitted to the average tissue expansion in $OsO_4$ solution (black dashed line, residual standard error $SE_{res}$ = 0.06%). (**g**) In situ X-ray-assisted staining of large tissue samples. The images show projections of a whole mouse brain incubated in 2% buffered $OsO_4$ at different points in time across 2.5 days. The arrowheads indicate the location of the staining front of accumulated osmium. Scale bar 2 mm.

The online version of this article includes the following source data and figure supplement(s) for figure 1:

**Source data 1.** Experimentally measured accumulation of heavy metals (blue traces) in buffered 2% $OsO_4$ solution in n=13 different samples.

**Source data 2.** Experimentally measured accumulation of heavy metals in a buffered solution of 2% OsO4 reduced with 2.5% potassium ferrocyanide K4[Fe(CN)6] (Ferro-redOs) in n=6 different samples.

**Source data 3.** Experimentally measured accumulation of heavy metals in buffered 2% Ferro-redOs solution with 5.56% formamide in n=4 different samples.

**Source data 4.** Experimentally measured accumulation of heavy metals in buffered 2% Ferro-redOs solution with 11.25% formamide in n=4 different samples.

**Source data 5.** Experimentally measured accumulation of heavy metals in a buffered solution of 2% OsO4 reduced with 2.5% potassium ferricyanide K3[Fe(CN)6] (Ferri-redOs) in n=2 different samples.

**Figure supplement 1.** As reported previously (*Mikula and Denk, 2015*, Figure 4b), X-ray imaging exhibits a contrast similar to scanning electron microscopy (SEM).

**Figure supplement 2.** Ultrastructural preservation at various tissue depths after 20 hr of X-ray-assisted staining with 2% buffered $OsO_4$.

**Figure supplement 3.** Spatial intensity profiles of accumulated heavy metals after t=20 hr of incubation (left column) and temporal tissue expansion profiles (right column) for different osmium reduction protocols in comparison to 2% buffered $OsO_4$ (blue, n=13) and buffered solution of 2% $OsO_4$ reduced with 2.5% potassium ferricyanide $K_4[Fe(CN)_6]$ (green, n=6) (mean ±s.d.).

# Results

We developed in situ time-lapsed X-ray-assisted staining with the goal of facilitating and accelerating the optimization of staining protocols for large biological samples such as whole mouse brains (*Abbott et al., 2020*). 4mm-brain punches of transcardially perfused mice were immersed in aldehyde fixatives for 36 hr. After washing the sample blocks with cacodylate buffer, the samples were immersed in staining solution and immediately placed in the acquisition chamber of a Zeiss Xradia 520 Versa 3D for X-ray microscopy (*Figure 1a*). The Xradia can be operated in two different modes:

- In the μCT mode, projections of the sample are acquired at different rotation angles in order to reconstruct a 3D computed tomograph of the sample. Depending on the required signal to noise and resolution, this mode allowed us to acquire approximately 1–2 computed tomographs per hour.
- In single-projection mode, a projection of the sample is acquired in a fixed position every few seconds without rotating the sample.

The advantage of the μCT mode is that arbitrary virtual reslices can be extracted to assess the staining progression in various parts of the sample. However, with our X-ray microscope, the temporal resolution in μCT mode was limited to 30–60 min per tomograph. Therefore we performed all quantitative experiments in single-projection mode where the temporal resolution for monitoring of the heavy metal diffusion and accumulation was on the order of a few seconds. Note, in this mode, the

pixel intensities of the resulting X-ray projection images are accumulated through the entire sample and therefore the correspondence to the pixel intensities in thin serial-section EM images is less accurate than in μCT mode (*Figure 1—figure supplement 1*). As soon as the samples got placed in the staining solutions, the heavy metals started accumulating in the immersed tissue. The stained tissue absorbs more X-rays than the unstained tissue, resulting in a noticeable intensity difference that can be used to track the diffusion and accumulation of the heavy metals in the tissue (*Figure 1b*).

## Quadratic scaling of incubation times with sample size

Osmium tetroxide ($OsO_4$) is one of the most commonly used contrast agents for lipid staining in electron microscopy due to its large atomic number and its ability to integrate into cellular membranes (*Porter et al., 1945*; *Palade, 1952*; *Watson, 1958*). The immersion times for $OsO_4$ vary between a few minutes to several days, depending on the sample size and tissue type, and are usually determined empirically (*Genoud et al., 2018*; *Hua et al., 2015*; *Tapia et al., 2012*; *Mikula and Denk, 2015*). However, the kinetics of $OsO_4$ staining of biological tissue are not well understood and so far had to be determined experimentally by trial and error. We therefore set out to measure the diffusion and accumulation of $OsO_4$ by placing 4 mm punches of aldehyde-fixed mouse brains in 2% buffered $OsO_4$ solution. The osmium diffused into the tissue and accumulated in the sample during the staining process (*Figure 1b*). The intensity of X-ray absorption in the projection images scales linearly with the local concentration/density of osmium (*Figure 1c*). The density of $OsO_4$ in the tissue increased beyond the density in the surrounding staining solution, indicating that the density of binding sites for $OsO_4$ in the tissue is higher than the concentration of the $OsO_4$ in the staining solution (*Figure 1b*). The diffusion and spatio-temporal accumulation of $OsO_4$ results in a staining front that propagates towards the center of the sample (*Figure 1b*). As expected for diffusive processes (*Carnevale et al., 1979*), the propagation of the $OsO_4$ staining front can be approximated by a quadratic model (*Figure 1d*). As in the case of other fixatives (*Medawar, 1941*), the osmium staining penetration in aldehyde fixed tissue obeys a quadratic scaling law, or rather a "rule of thumb", for how the necessary incubation time t depends on the staining depth or sample size x:

$$t \propto x^2 \tag{1}$$

This means, for example, that if one would adapt an established $OsO_4$ staining protocol for 3 X larger samples, one would have to prolong the incubation time by 9 X in order to produce comparable staining results. Similarly, the time it takes for any point in the sample to reach a given concentration is proportional to the square of its distance to the sample surface (*Carnevale et al., 1979*).

## Monitoring staining kinetics and tissue deformation

After 20 hr of incubation the staining density of osmium is homogeneous across the first 1000 μm of cortical depth (*Figure 1e*). In addition, the ultrastructure is well preserved (*Figure 1—figure supplement 2*). Reduced osmium is another commonly used staining agent that is known to result in higher contrast for electron microscopy images than non-reduced $OsO_4$. Typically, a buffered 2% $OsO_4$ solution is reduced with 2.5% potassium ferrocyanide ($K_4[Fe(CN)_6]$) (*Hua et al., 2015*; *Willingham and Rutherford, 1984*; *Mikula and Denk, 2015*), which we will call Ferro-redOs throughout this manuscript. Consistent with previous reports on Ferro-redOs staining in large samples (*Hua et al., 2015*; *Mikula and Denk, 2015*) we found that for Ferro-redOs the accumulation of heavy metals peaks at a depth between 300 and 800 μm (*Figure 1e*), whereas the tissue above or below that depth is stained less. Traditionally, this band of more heavily stained tissue has been associated with precipitated osmium that hinders the diffusion and prevents homogenous staining (*Genoud et al., 2018*; *Hua et al., 2015*; *Mikula and Denk, 2015*), in particular deeper in the tissue. No such band is present in tissue immersed in non-reduced $OsO_4$. (*Mikula and Denk, 2015*) reported that homogeneous staining of large samples with Ferro-redOs was achieved by adding formamide to the Ferro-redOs solution. However, the underlying mechanisms by which formamide acts are not known (*Mikula and Denk, 2015*; *Genoud et al., 2018*). (*Mikula and Denk, 2015*) hypothesized that formamide might prevent precipitation by generally solubilizing compounds or that it might allow highly charged molecules to cross membranes more easily. Adding formamide to the Ferro-redOs solution indeed resulted in more homogeneous staining (*Figure 1e*, ), but the heavy metal density was lower than in osmium only. (*Mikula and Denk, 2015*) reported that the tissue tends to expand at high concentrations of

formamide (>50%). But we found that the tissue expands even at lower formamide concentrations, while the amount of expansion depends on both the formamide concentration and the incubation duration (*Figure 1f*). For a concentration of 11.25% formamide, the sample height increased by about 15% within 20 hr of incubation, whereas for a formamide concentration of 5.56%, the tissue height expanded only by approximately 10% (*Figure 1f*). Note, in the first 3 hr of incubation in Ferro-redOs with 5.56% formamide the tissue actually shrank, suggesting that there are opposite forces acting on the tissue. Indeed, in Ferro-redOs solution without any formamide, the tissue shrank by about 5% in height (*Figure 1f*). In contrast, we found that in 2% buffered $OsO_4$ solution the sample height expands by about 5% (*Figure 1f*). However, we found no sample expansion if the reduced osmium solution was prepared with 2.5% potassium ferricyanide $K_3[Fe(CN)_6]$+ 8 a,b), another commonly used reducing agent for $OsO_4$.

*Hua et al., 2015* suggested an alternative approach to achieve homogeneous staining in 1 mm brain punches with Ferro-redOs without formamide, in which osmium and the reducing agent are applied separately. First, the samples are immersed in buffered 2% $OsO_4$ solution for 90 min. Subsequently, the samples are placed for 90 min in buffered 2.5% $K_4[Fe(CN)_6]$ without any washing step in-between. We repeated this procedure for 4 mm brain punches: First, the samples were incubated in 2% buffered $OsO_4$ for 22 hours resulting in homogeneous staining throughout the sample. Next, we placed the sample directly without any washing step in 2.5% buffered $K_4[Fe(CN)_6]$ solution. (*Hua et al., 2015*) hypothesized that the main effect of reducing agents such as $K_4[Fe(CN)_6]$ is to convert VIII-oxidized water-soluble osmium into an VI-oxidized water-soluble form which is thought to generate additional, non-polar $OsO_2$ to be deposited in the membrane increasing the heavy metal content and the contrast in electron microscopy. However, we found that the heavy metal density in the $K_4[Fe(CN)_6]$ immersed samples did not increase, but rather decrease from the sample surface towards the center as time progressed (*Figure 2a*). This suggests that $K_4[Fe(CN)_6]$ removes or "washes out" osmium from the sample (*Litman and Barrnett, 1972*) resulting in an inverse staining gradient. This "washing effect" is stronger for longer incubation times as well as close to the sample surface and around blood vessels (anecdotal observation in electron microscopy images, data not shown). For incubation times longer than 12 hr, $K_4[Fe(CN)_6]$ readily dissolves and disintegrates the tissue (*Figure 2—figure supplement 1*). Interestingly, no 'washing effect' was observed if the same procedure was repeated with 2.5% potassium ferricyanide $K_3[Fe(CN)_6]$, but the sample height shrank by about 2% (*Figure 1—figure supplement 3*,f, ). Similarly, only a small reduction in heavy metal density could be observed when the sample was washed with double-distilled $H_2O$ for 20 hr (*Figure 2—figure supplements 2 and 3*).

## The kinetics of osmium tetroxide staining

The chemistry and diffusion-reaction-advection kinetics of $OsO_4$ staining in aldehyde-fixed tissue are not well understood. In brain tissue, $OsO_4$ is thought to mainly react with carbon-carbon double bonds, in particular of unsaturated lipids, but it has also been reported to react with some amino acids and nucleotides (*Khan et al., 1961*; *Korn, 1967*; *White et al., 1976*; *Schroeder, 1980*). At the beginning of the $OsO_4$ incubation, the density of $OsO_4$ steeply increases and peaks close to the surface of the sample. As the incubation time progresses, the $OsO_4$ continues to diffuse and accumulates deeper in the tissue (*Figure 2b, d and e*). Due to the curved surface of the cortical tissue sample, the projected tissue thickness is lower (*Figure 2f*) and there are fewer binding sites for osmium, which results in a lower density of osmium closer to the sample surface (*Figure 2b*). At any given time, the accumulation seems to plateau from the surface towards the center of the sample, indicating that the binding of $OsO_4$ saturates (*Figure 2d and e*). However, we found that the saturated density in the samples slowly increased over time from about 129 nmol/mm$^3$ after 1 hr of incubation to 160 nmol/mm$^3$ after 10 hr of incubation (*Figure 2b, d and e*), while the density of the solution remained stable at about 82 nmol/mm$^3$. This suggests that additional binding sites for $OsO_4$ become available slowly over time. The reactions underlying the 'unmasking' of binding sites are not known, but we hypothesize that it depends at least partially on the availability and the local concentration of unbound, freely diffusing osmium. Hence, at the macroscale level, the accumulation of osmium in biological tissue can be described by four coupled processes (*Figure 2g*): As the osmium diffuses passively into the tissue, it binds to the available binding sites in the tissue and additional binding sites are slowly being unmasked, while the tissue expands by around 5% within 20 hr of incubation (*Figure 1f*). The kinetics

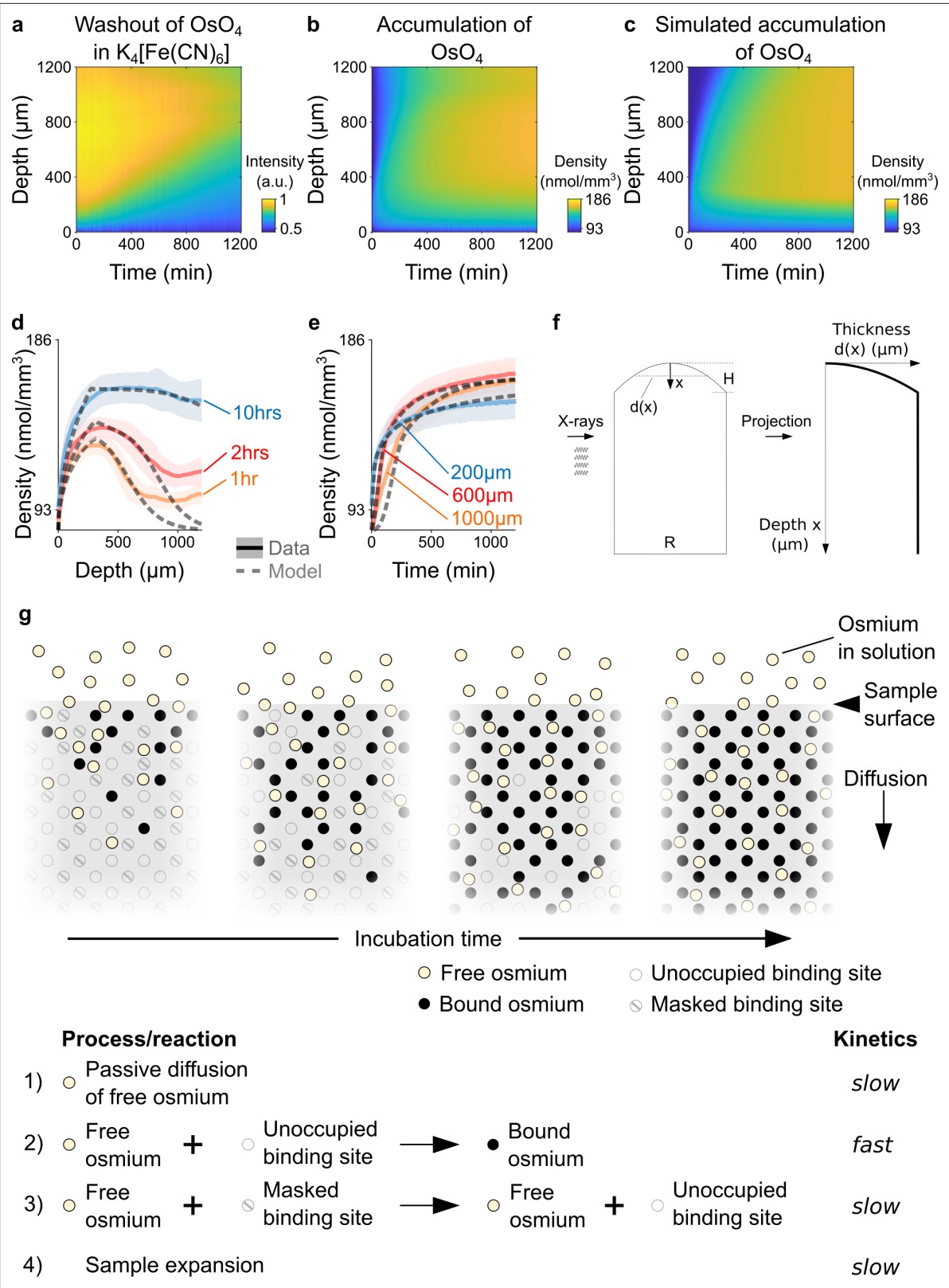

**Figure 2.** Kinetics and tissue mechanics of electron microscopy staining. (**a**) Spatio-temporal washout of osmium in samples incubated in 2.5% potassium ferrocyanide $K_4[Fe(CN)_6]$ (n=6). The samples have been stained with 2% $OsO_4$ for 22 hrs prior to placing them in $K_4[Fe(CN)_6]$. The osmium gets washed out from the sample surface towards the center as the incubation time increases. (**b**) Average experimentally measured spatio-temporal osmium density ($nmol/mm^3$) accumulation in samples (n=13) immersed in 2% buffered $OsO_4$ solution. Because of the surface curvature of the cortical

*Figure 2 continued*

samples, the tissue thickness decreases towards the surface (x=0) and results in a lower intensity in the projection images (see tissue geometry model in f). (**c**) Simulated spatio-temporal osmium density (nmol/mm$^3$) accumulation fitted to the average experimental data in b. (**d**) Density profiles (nmol/mm$^3$) of the spatial accumulation of osmium (n=13, mean ±s.d.) after 1 hr (orange), 2 hrs (red), and 10 hr (blue) of incubation in 2% buffered OsO$_4$ overlaid with the simulation of the diffusion-reaction-advection model (dashed lines). (**e**) Density profiles (nmol/mm$^3$) of the temporal accumulation of osmium (n=13, mean ±s.d.) at a depth of 200 μm (blue), 600 μm (red), and 1000 μm (orange) during incubation in 2% buffered OsO$_4$, overlaid with the simulation of the diffusion-reaction-advection model (dashed lines). (**f**) The geometry of the 4 mm brain punches was modeled as a cylinder with a curved surface. The curvature of the sample surface is approximated by a circle with radius $\frac{1}{2H} \cdot \left( H^2 + \frac{1}{4}R^2 \right)$, where $R$ is diameter of the cylinder and $H$ is the height of the curved surface. The projected tissue thickness as a function of the depth x from the sample surface is then given by $d\left( x \right)$: $d\left( x \right) = 2 \cdot \sqrt{\frac{x}{H} \cdot \left( H^2 + \frac{1}{4}R^2 \right) - x^2}$, for $0 \leq x \leq H$ and $d\left( x \right) = R$, for $x > H$ (**g**) Diffusion-reaction-advection model for osmium staining. The model combines four coupled processes: (1) Free OsO$_4$ diffuses passively into the tissue and (2) binds to the available binding sites. (3) In the presence of freely diffusing OsO$_4$ previously masked binding sites are slowly turned into additional binding sites for OsO$_4$. (4) The sample expands by about 5% in sample height within 20 hr of incubation (see *Figure 1f*).

The online version of this article includes the following source data and figure supplement(s) for figure 2:

**Source data 1.** Experimentally measured washout of heavy metals in buffered 2.5% potassium ferrocyanide K$_4$[Fe(CN)$_6$] solution in n=6 different samples that have previously been stained with buffered OsO$_4$ for 22 hr.

**Source data 2.** Experimentally measured washout of heavy metals in buffered 2.5% potassium ferricyanide K3[Fe(CN)6] solution in n=4 different samples that have previously been stained with buffered OsO4 for 22 hrs.

**Figure supplement 1.** Long incubation in potassium ferrocyanide dissolves and disintegrates the tissue.

**Figure supplement 2.** Experimentally measured change in heavy metal accumulation of a sample immersed in double distilled H$_2$O (ddH$_2$O).

**Figure supplement 3.** (a) Spatial intensity profile of accumulated heavy metals at the beginning t=0 hr and after t=20 hr of immersion in double distilled H$_2$O (n=1).

**Figure supplement 4.** Effective diffusion coefficient D$^e$, unmasking rate constant k$^{unmask}$, binding rate constant k$^{on}$, the OsO$_4$ concentration in the solution C$^0$, the initial density of binding sites S$^0$, the initial density of masked binding sites M$^0$ and the sample curvature height H for n=13 different 4 mm punches that have been immersed for 22 hr in 2% buffered OsO$_4$ solution.

**Figure supplement 5.** Experimentally measured accumulation of heavy metals (blue traces) in buffered 2% OsO$_4$ solution of a 4 mm punch of aldehyde-fixed liver tissue (n=1).

---

of these four processes can be described with the following coupled (non-linear) partial differential equations:

$$\frac{\partial f(x,t)}{\partial t} = D^e \cdot \frac{\partial^2 f(x,t)}{\partial x^2} - k^{on} \cdot s(x,t) \cdot f(x,t) \quad - \frac{\partial}{\partial x}\left[ f\left( x,t \right) \cdot v\left( x,t \right) \right] \tag{2}$$

$$\frac{\partial b(x,t)}{\partial t} = k^{on} \cdot s(x,t) \cdot f(x,t) \quad - \frac{\partial}{\partial x}\left[ b\left( x,t \right) \cdot v\left( x,t \right) \right] \tag{3}$$

$$\frac{\partial s(x,t)}{\partial t} = -k^{on} \cdot s(x,t) \cdot f(x,t) + k^{unmask} \cdot m(x,t) \cdot f(x,t) \quad - \frac{\partial}{\partial x}\left[ s\left( x,t \right) \cdot v\left( x,t \right) \right] \tag{4}$$

$$\frac{\partial m(x,t)}{\partial t} = -k^{unmask} \cdot m(x,t) \cdot f(x,t) \quad - \frac{\partial}{\partial x}\left[ m\left( x,t \right) \cdot v\left( x,t \right) \right] \tag{5}$$

f(x,t) is the density of freely diffusing OsO$_4$, b(x,t) is the density of osmium bound to the tissue, s(x,t) is the density of available binding sites and m(x,t) corresponds to the density of masked binding sites at time t and a radial depth x from the cortical surface. D$^e$ is the effective diffusion coefficient for OsO$_4$, k$^{on}$ is the reaction rate constant for the binding of osmium to the tissue and k$^{unmask}$ is the reaction rate constant at which masked binding sites get unmasked. In addition, the local change of the tissue geometry, e.g. the expansion as observed for OsO$_4$, results in a velocity field v(x,t) that describes the local spatio-temporal flow of the density of each component c(x,t):

$$\frac{\partial}{\partial x}\left[ c\left( x,t \right) \cdot v\left( x,t \right) \right] = \frac{\partial c(x,t)}{\partial x} \cdot v\left( x,t \right) + c\left( x,t \right) \cdot \frac{\partial v(x,t)}{\partial x} \tag{6}$$

The flow consists of an advection term $\frac{\partial c(x,t)}{\partial x} \cdot v\left( x,t \right)$, corresponding to elemental volumes moving with the local flow, and a dilution term $c\left( x,t \right) \cdot \frac{\partial v(x,t)}{\partial x}$ that describes the local volume change. While we assume that the tissue density is homogeneous across the cortical thickness, we consider differences in the sample thickness due to the curvature of the cortical surface. Additional information on the model assumptions can be found in the methods section.

We fitted the parameters of the system of reaction-diffusion-advection equations for a tissue sample with the projection geometry shown in *Figure 2f* to the average measured spatio-temporal

accumulation of OsO$_4$ (*Figure 2b–e*) as well as to the individual samples (n=13, ). The fitted parameters of the diffusion-reaction-advection model (*Figure 2—figure supplement 4*) capture the measured diffusion-reaction-advection kinetics of osmium staining well (*Figure 2c*; residual standard error SE$_{res}$ = 0.026 ± 0.005, mean ±s.d.).

## Discussion

In situ X-ray-assisted staining is a new tool for monitoring the different steps of *en bloc* staining protocols for electron microscopy. By opening the 'black box' of staining protocols, X-ray-assisted staining enables, for the first time, to observe directly the diffusion of heavy metals and their microscopic effects on the tissue. While the technique will be useful to develop novel staining protocols for small to intermediate sized samples with dimensions less than 1 mm, it will particularly accelerate and facilitate the development and optimization of new staining protocols for large biological samples such as whole brains (*Figure 1g*).

We monitored the accumulation of osmium in different commonly used staining solutions and found significant differences in the accumulation densities. In addition, we found that the tissue expands and shrinks significantly depending on the composition of the staining solution. In particular, we found that the tissue height increased by 5% in buffered osmium solution, while the tissue height decreased by 5% in buffered Ferro-redOs solution. We hypothesize that this shrinkage could provide a mechanistic explanation for the long standing issue of diffusion barrier formation in large samples during Ferro-redOs incubation.

The method also revealed that K$_4$[Fe(CN)$_6$], a reducing agent that was believed to enhance the accumulation of osmium in membranes, in contrast can wash out osmium from stained tissue if applied separately. If K$_4$[Fe(CN)$_6$] removes osmium from the sample, why does the protocol modification of *Hua et al., 2015* result in enhanced electron microscopy contrast for samples with dimensions of up to 1 mm? A potential explanation could be that K$_4$[Fe(CN)$_6$] clears the cytosol more efficiently than the cell membranes. As a consequence, the osmium content in the cytosol gets washed out more/faster, which might result in an overall electron microscopic contrast increase between the membranes and the cytosol. However, the chemical mechanisms by which K$_4$[Fe(CN)$_6$] interact with the tissue remain poorly understood and shining a light on those processes will require additional work.

X-ray assisted staining opens the door for in silico staining protocol optimization, by enabling to measure important heavy metal staining parameters such as the effective diffusion coefficients and reaction rate constants directly in situ. We derived a diffusion-reaction-advection model that describes the accumulation of osmium in homogeneous tissue by four distinct, but coupled, processes. Osmium diffuses passively into the tissue and binds there quickly to available binding sites, for example unsaturated lipids. Thereby the tissue expands by about 5% in height. Additional binding sites are being unmasked over the course of hours, which enables the tissue to accumulate additional osmium. While the binding of osmium appears to be nearly instantaneous, the passive diffusion and unmasking reactions are slow and appear to be rate-limiting for the staining. The mechanisms and reactions underlying the unmasking of additional binding sites for osmium are not known. Unsaturated lipids, one of the prominent binding sites for osmium, can be masked by lipid-protein complexes. In the presence of osmium, these complexes could dissociate and make the previously masked unsaturated lipid available for binding osmium (*Clayton, 1959*; *Lauder and Beynon, 1993*; *Ashhurst, 1961*; *Wigglesworth, 1975*). A recent study used time-of-flight secondary ion mass spectrometry to analyze the colocalization of different fatty acid species with osmium at the tissue surface (*Belazi et al., 2009*). As expected, the unsaturated fatty acids colocalize well with the tissue-bound osmium. However, they also report a remarkable complementary clustering of saturated and unsaturated fatty acids in osmium stained tissue that appears not to be present in unstained tissue. Hence, it could be that osmium can convert saturated lipids to unsaturated lipids through hydrogenation/oxidation. Alternatively, the masked binding sites could also be interpreted as a different binding partner that reacts much slower with osmium.

Biological samples such as brains are non-homogeneous media. They are composed of different tissue and cell types and therefore the diffusion of staining agents is expected to be heterogeneous. Nevertheless, the fitted model parameters reproduce the observed kinetics of osmium accumulation relatively well up to a depth of about 700–800 μm from the surface. Deeper in the tissue osmium that diffused from secondary sources such as ventricles or axon tracts appear to contribute to the

accumulation of osmium. For example, we found that white matter and myelinated axon tracks tend to accumulate osmium faster and stronger compared to gray matter (, ). Such secondary sources and tissue inhomogeneities are not considered in our model. Still, the fitted model parameters appear consistent with comparable values from the literature, although we also find some discrepancies. For example, in mouse brains we expect to find about 117.975 nmol/mm$^3$ double-bonds on unsaturated lipids (see methods for the derivation). Because the dominant binding reaction for osmium is assumed to be the formation of mono- and di-ester bonds with unsaturated lipids which has been described to result in the deposition of $OsO_2$, the density of double bonds should roughly correspond to the density of available binding sites. But the proposed model predicts 2.999 times more binding sites, even without unmasking. One interpretation would be that unsaturated lipids are not the dominant reaction partner and that other reaction partners such particular amino acids or nucleotides account for about two third of the available binding sites. Alternatively, this could indicate that each unsaturated lipid double bond gets associated with on average 3 osmium atoms, for example through $Os_3O_5^-$ and $Os_3O_5H^-$ isotopes that have previously reported to exist in osmium-stained adipose mouse tissue (*Belazi et al., 2009*). Similarly, it has been reported that if the mono- and di-ester products are left standing in solution, a highly insoluble trimer osmium product is formed (*Schroeder, 1980*).

We expect that X-ray-assisted staining will be very useful in establishing a more refined picture of the heterogeneity of the heavy metal diffusion processes in different tissue types and compartments as well as in different conditions. For example, we also measured and modeled the accumulation of osmium in liver tissue (*Figure 2—figure supplement 5*). It will also be interesting to examine how changes in the extracellular space and different fixation protocols or microwave-assisted incubation affect the diffusion dynamics (*Korogod et al., 2015*; *Fulton and Briggman, 2020*; *Jensen and Harris, 1989*; *Login and Dvorak, 1988*).

Furthermore, X-ray-assisted staining can be used at higher resolution and in combination with computed tomography for producing a more detailed picture of the tissue mechanics involved in different staining steps. We and several other labs have noticed that brain tissue expands significantly during incubation in 'osmium amplifiers' such as the osmiophilic thiocarbohydrazide (TCH) (*Seligman et al., 1966*). (*Mikula and Denk, 2015*) reported that nitrogen bubbles are formed during TCH incubation that could be responsible for the formation of tissue cracks in large samples. X-ray-assisted staining could be used to detect and monitor the formation of tissue cracks and to explore ways to reduce these artifacts in large samples.

Aside from accelerating the development of novel staining protocols this new method will also allow for improved closed-loop quality control of the staining process. Large samples tend to be biologically more variable and more difficult to precisely fine-tune all the parameters in the staining protocol ahead of time to account for these individual differences. Hence, X-ray-assisted staining can be used to adjust the parameters in real time to compensate for these individual differences. This is particularly important for experiments in which the sample yield is low or the individual sample is precious, as for example in the case of human brains or functionally imaged brains of behaving animals that require several months of behavioral training. However, this will require calibrating what is measured with X-ray microscopy to the corresponding appearance at an electron microscopic level, because the image formation process and the resulting contrast, signal-to-noise, etc. for both imaging modalities depend not only on the staining protocol, but also on the imaging modality, energy, flux, etc. (*Du and Jacobsen, 2018*).

## Methods
### Sample preparation
Animal use procedures were approved by the Princeton University Institutional Animal Care and Use Committee (protocol number 2000) and carried out in accordance with National Institutes of Health standards (AAALAC International Institutional Number: Unit #1001, PHS assurance ID D16-00273). The experiments were carried out with C57BL/6 J mice (Jackson Laboratories, Bar Harbor, Maine, USA) of both sexes, age ranging from 13 to 58 weeks (mean ± sd: 27±9 weeks). Prior to tissue sample extraction, the mice were anesthetized by isoflurane inhalation (4%) and euthanized with an intraperitoneal injection of a ketamine (100 mg/kg) and xylazine (10 mg/kg) overdose. Next, the animals were perfused transcardially with ≥50 ml fixative solution (1.3–1.5% glutaraldehyde (GA) and 2.5–2.6%

paraformaldehyde (PFA) in 0.14–0.15 M cacodylate buffer with 2.0–2.1 mM $CaCl_2$ at pH 7.4; (GA, PFA) and cacodylate buffer Electron Microscopy Sciences, Hatfield, PA; $CaCl_2$, Sigma Aldrich, St. Louis, MO). After perfusion, the animal was left for 15–60 min on ice. Subsequently, the brain was removed in order to extract tissue samples at the center of the cortical surface of each hemisphere using biopsy punches of 4 mm diameter. The liver tissue was extracted with a 4 mm biopsy punch from the center of the liver after perfusion. The samples were kept in fixative solution at 4 °C for 36 hr (4 mm biopsy punches) or for 12 hr (whole mouse brain). Next, the samples were washed 6–7 times for 30 min in 0.15 M cacodylate buffer with 2 mM $CaCl_2$.

Glass vials were prepared prior to the experiments with a 1–2 mm layer of Sylgard 184 (Electron Microscopy Sciences, Hatfield, PA) at the bottom. The whole mouse brain was glued with a small drop of Vetbond tissue adhesive (3 M, Hanna Pharmaceutical Supply Co., Inc, Wilmington, DE) to the bottom of the vial. For the 4 mm biopsy punch samples, a hole with a diameter of 4 mm was cut out of the Sylgard layer. Next, the samples were placed into the sylgard hole and were glued to the bottom of the glass vial with a small drop of Vetbond tissue adhesive. Subsequently, the samples were immersed in fresh cacodylate buffer solution for the transfer to the X-ray microscope. At the microscope, presets (see X-ray microscopy section below) were loaded for image acquisition. After that the washing solution was replaced with the staining solution and the sample was placed into the X-ray microscope. The acquisition of X-ray images was typically started within 5.66±2.35 min. (mean ±s.d.) after adding the staining solution. The following staining solutions have been assessed at room temperature for this study (all buffered with 0.15 M cacodylate buffer and 2 mM $CaCl_2$):

- Osmium tetroxide: 1%, 2% and 3% $OsO_4$ (prepared from 2 mL 4% osmium tetroxide solution, Electron Microscopy Sciences, Hatfield, PA)
- Potassium ferrocyanide: 2.5% potassium ferrocyanide (prepared from crystalline 25 g potassium hexacyanoferrate(II) trihydrate powder, Sigma Aldrich, St. Louis, MO)
- Ferro-redOs: 2% $OsO_4$, 2.5% potassium ferrocyanide
- Ferro-redOs in 11.25% formamide: 2% $OsO_4$, 2.5% potassium ferrocyanide, 11.25% formamide (prepared from 25 mL formamide solution, Sigma Aldrich, St. Louis, MO)
- Ferro-redOs in 5.56% formamide: 2% $OsO_4$, 2.5% potassium ferrocyanide, 5.56% formamide
- Potassium ferricyanide: 2.5% potassium ferricyanide (EMS #13746-66-2)
- Potassium ferricyanide reduced osmium: 2% $OsO_4$, 2.5% potassium ferricyanide

## Sample embedding, sectioning, and electron microscopy

After staining, the samples were washed with double-distilled $H_2O$ 3 times for 30 min. Next, the tissue was dehydrated with iteratively increasing concentrations of ethanol (EtOH, 200 proof, anhydrous,≥99.5%, Sigma Aldrich, St. Louis, MO) in double-distilled $H_2O$ for 30 min per step: 25% EtOH, 50% EtOH, 75% EtOH, 90% EtOH, 95% EtOH, 100% EtOH, 100% EtOH. Subsequently, the samples were transferred to 100% acetonitrile (ACS grade, Electron Microscopy Sciences, Hatfield, PA) for two 30-min washes. The tissue was embedded in hard Epon in an iterative concentration series: 1:2 Epon: acetonitrile for 12 hr, 1:1 Epon: acetonitrile for 24 hr, 2:1 Epon: acetonitrile for 2 days with fresh resin every 8–12 hr. Finally, the sample was left on the bench at room temperature for 6 days, with fresh resin every 8–12 hr, before it was placed in a 60 °C oven for curing for 5–7 days.

Serial sections of ~40 nm thickness were collected using an UC7 ultramicrotome from Leica. To assess the ultrastructural preservation of the tissue, the sections were imaged with a Zeiss Sigma SEM (Zeiss Microscopy, Oberkochen, Germany) with a backscattered electron detector (BSD) and the following acquisition parameters: 38 µs dwell time, 4 nm pixel size, probe current 200 nA, EHT voltage 6 kV, BSD Gain = High, all sectors of the detector ON.

## X-ray microscopy

All X-ray microscopy experiments were performed on a Zeiss Xradia 520 Versa 3D X-ray microscope (Zeiss, Thornwood, NY). The samples were immersed in heavy metal solution in glass containers tightly sealed with Parafilm (Bemis Company, Inc, Neenah, WI) just before being placed in the recording chamber of the Xradia. Test images were acquired to check imaging parameters. For the biopsy punch samples, projections at a fixed view angle (rotation angle step size = 0°) were taken every 22–60 s resulting in about 1300–1350 images per sample over 22 hr of continuous acquisition. For the whole mouse brain, projections were taken every 22–60 s and after each projection the sample was rotated

by 1°. All images were acquired using the image acquisition software Scout-and-Scan Control System (Zeiss, Thornwood, NY) with the following parameters:

- Objective: 0.4 X
- Tube Voltage = 100 kV
- Output power = 7 W
- Source filter = air (no filter)
- Exposure time = 20 s.
- Effective pixel size = 2–3 µm (biopsy punches) and 13–14 µm (whole mouse brain)

## Image preprocessing

First, the individual images and metadata of each projection of each sample were extracted from the proprietary *.txrm and *.xrm files as saved by the Xradia image acquisition software. In addition, a time stamp of the acquisition time point was extracted from the image metadata for each projection. A custom cross-correlation based image alignment procedure (*Wanner et al., 2016*) was used to correct for translational offsets between images of subsequent time points. The projection images were flat-field corrected to account for radial intensity variations due to the cone-shaped X-ray beam as follows:

1. For each sample, a median image $im_{median}$ was calculated from the 2nd to 5th projection images right after acquisition onset.
2. The following quadratic 2d function was fitted to $log(im_{median})$ using the built-in function *lsqcurvefit* in MATLAB:
$$Z(x,y) = b_5 \cdot x^2 + b_4 \cdot y^2 + b_3 \cdot x + b_2 \cdot y + b_1$$
3. At each time t, the projection image was flat-field corrected using $Z(x,y)$ with the fitted parameters:
$$\hat{im}_t(x,y) = \frac{im_t(x,y)}{exp(Z(x,y))} \cdot max(exp(Z))$$

## Measuring the X-ray absorption and local heavy metal density

The pixel intensities of the detected absorbed X-rays were used as a proxy for the density of heavy metal accumulation in the tissue. In each sample, we quantified the pixel intensities in each projection taken at time t along a cross-section that pointed from the cortical surface radially towards the white matter (*Figure 1b*). The cross-sections were manually selected such that they started at a well detectable surface boundary and they were located at least 1 mm away from the vertical sample borders. For each sample i, the intensity $w_i(x,t)$ of the transmitted X-rays was quantified by averaging the pixel intensity of the cross-section at a depth $x$ at time $t$ within a width of ±25 pixels perpendicular to the axis of the cross-section. x=0 corresponds to the cortical surface of the brain samples or the surface of the liver for the liver samples. The intensity of X-ray absorption was defined as $u_i(x,t) = 10,000 - w_i(x,t)$. For each time point t, the baseline intensity $b_i(t)$ was calculated by averaging the pixels outside of the sample within –50 µm and –150 µm from the sample surface along the cross-section within a width of ±25 pixels perpendicular to the axis of the cross-section. To account for baseline fluctuation in the staining solution and the X-ray illumination the intensity of X-ray absorption was baseline-corrected by $u_i^b(x,t) = u_i(x,t) - (b_i(t) - median(b_i))$.

To standardize $u_i^b(x,t)$, the following quantities where calculated for each sample:

1. $\underline{b_i} = mean\left(u_i^b(x,t)\right)$ was calculated for all pixels along the cross-section outside of the sample within 0 µm and –200 µm from the sample surface.
2. $u_i^{min} = \underline{b_i} - \frac{1-0.9}{0.9}(m - \underline{b_i})$
3. $u_i^{ref} = u_i^{min} - a \cdot \left(m - u_i^{min}\right)$
4. $u_i^{norm} = (1 + a) \cdot \left(m - u_i^{min}\right)$

Finally, $u_i^b(x,t)$ was standardized to $u_i(x,t) = \left(u_i^b(x,t) - u_i^{ref}\right)/u_i^{norm}$.

This procedure ensures that the origin is at 0 for the linear relationship between the X-ray absorption intensities in the projection images and the different $OsO_4$ concentrations as shown in *Figure 1c* and that a pixel intensity of 0.9 roughly corresponds to the saturated staining in 2% $OsO_4$. For this, the factors $a = 0.61749$ and $m = 6420$ were determined empirically.

## Measuring the linear relationship between X-ray absorption and OsO$_4$ concentration

The X-ray absorption/pixel intensity of cacodylate-buffered solutions with 1%, 2% and 3% OsO$_4$ was measured the same way as for the X-ray absorption in stained tissue.

## Monitoring of tissue expansion and shrinkage

Intensity based heuristics were used in a semi-automated procedure to detect the cortical surface in each image of each sample. The glass vial bottom was used as a reference point to measure the distance to the cortical surface $H(t)$ at time $t$. Right after placing the samples in the X-ray the height of the samples was $H_0 = H(t = 0)$ and the change in sample height (expansion or shrinkage) was quantified as $\frac{H(t) - H_0}{H_0} \cdot 100\%$.

## Tissue expansion modeling

The expansion of the tissue in buffered OsO$_4$ solution was modeled by the monomolecular, saturating growth function $g(t) = a_1 - a_2 \cdot exp\left(-\frac{t}{\tau}\right)$. The built-in function *lsqcurvefit* of MATLAB was used for parameter fitting. The fitted parameters for the average expansion in **Figure 1f** were $a_1 = 5.8697\%$, $a_2 = 5.6924\%$, $\tau = 588.23$ *min*. Under the assumption that the expansion of the tissue domain in buffered osmium solution is isotropic, the corresponding velocity field v(x,t) of the flow in **Equation 6** can be characterized by:

$$v(x,t) = -x \cdot \frac{\partial(1 + g(t)/100)}{\partial t} = -x \cdot \frac{a_2}{100 \cdot \tau} \cdot exp\left(-\frac{t}{\tau}\right).$$

## Tissue projection geometry model

While we assumed that the tissue density is homogeneous across the brain punch, we took into account the variation in the tissue thickness of the 4 mm brain punch. **Figure 2f** shows a sketch of the modeled tissue geometry. The geometry of the 4 mm brain punches is modeled as a cylinder with a curved surface. The diameter of the cylinder is $R$ and the curvature of the sample surface is approximated by a circle with radius $\frac{1}{2H} \cdot \left(H^2 + \frac{1}{4}R^2\right)$. The thickness of the sample as a function of the depth x from the sample surface is then given by $d(x)$:

$$d(x) = 2 \cdot \sqrt{\frac{x}{H} \cdot \left(H^2 + \frac{1}{4}R^2\right) - x^2} \text{ , for } 0 \leq x \leq H \text{ and } d(x) = R, \text{ for } x > H.$$

## Diffusion-reaction-advection modeling

The diffusion-reaction-advection model described by **Equations 2–5** makes the following assumptions:

- the cortical surface is in contact with an infinite source of staining agents at constant concentration,
- the composition of the 4 mm brain punches can be approximated by a semi-infinite, homogeneous medium,
- the tissue thickness is given by $d(x) = 2 \cdot \sqrt{\frac{x}{H} \cdot \left(H^2 + \frac{1}{4}R^2\right) - x^2}$ , for $0 \leq x \leq H$ and $d(x) = R$, for $x > H$
- the kinetics of OsO$_4$ diffusion can be described by Fick's second law,
- the binding of OsO$_4$ to the tissue is irreversible and can be described by first-order reaction kinetics
- the density of available OsO$_4$ binding sites is limited,
- the unmasking of additional binding sites is irreversible, follows first-order reaction kinetics and depends on the local concentration of free OsO$_4$
- the tissue expands isotropically and can be modeled by a monomolecular, saturating growth function, which induces a local velocity field $v(x,t) = -x \cdot \frac{\partial(1 + g(t)/100)}{\partial t} = -x \cdot \frac{a_2}{100 \cdot \tau} \cdot exp\left(-\frac{t}{\tau}\right)$.

Because the projected intensity also includes any X-ray absorbing components surrounding the sample, we have to consider the change in free osmium density outside of the sample $o(x,t)$ for fitting the model to the experimental data:

$$\frac{\partial o(x,t)}{\partial t} = -\frac{\partial}{\partial x}\left[o(x,t) \cdot v(x,t)\right] \tag{7}$$

The initial conditions (t=0) for the system of coupled partial differential *Equations 2–5; 7* were chosen as follows:

- concentration of free $OsO_4$ in the sample: $f(x \leq 0, t = 0) = C^0 \cdot R$ and $f(x > 0, t = 0) = 0$
- density of bound $OsO_4$: $b(x, t = 0) = 0$
- density of available binding sites: $s(x \leq 0, t = 0) = 0$ and $s(x > 0, t = 0) = S^0 \cdot d(x)$
- density of masked binding sites: $m(x \leq 0, t = 0) = 0$ and $m(x > 0, t = 0) = M^0 \cdot d(x)$
- concentration of free $OsO_4$ surrounding the sample: $o(x \leq 0, t = 0) = C^0 \cdot (D - R)$ and $o(x > 0, t = 0) = C^0 \cdot (D - d(x))$, where $D = 25mm$ is the vial diameter.

The boundary conditions of the system of coupled partial differential *Equations 2–5; 7* for $x \in [0, L]$ with maximal depth $L = 3mm$ were chosen as follows:

- concentration of free $OsO_4$: $f(x = 0, t) = C^0 \cdot R$ and $f(x = L, t) = 0$
- density of bound $OsO_4$: $b(x = 0, t) = 0$ and $b(x = L, t) = B^0 \cdot d(L)$
- density of available binding sites: $s(x = 0, t) = 0$ and $s(x = L, t) = S^0 \cdot d(L)$
- density of masked binding sites: $m(x = 0, t) = 0$ and $m(x = L, t) = M^0 \cdot d(L)$
- concentration of free $OsO_4$ surrounding the sample: $o(x = 0, t) = C^0 \cdot (D - R)$ and $o(x = L, t) = C^0 \cdot (D - d(L))$
- No flux beyond $x = L$: $\frac{\partial f(x=L,t)}{\partial x} = \frac{\partial s(x=L,t)}{\partial x} = \frac{\partial m(x=L,t)}{\partial x} = \frac{\partial o(x=L,t)}{\partial x} = 0$

The MATLAB function *pdepe* was used to solve/simulate the initial-boundary value problem described above.

## Diffusion-reaction-advection model fitting

$C^0$ was calculated by averaging $u_i^b(x, t)$ over $x \in [-150, -50]$ μm from the sample surface (i.e. outside of the sample) and $t \in [0, 1200]$ min and dividing it by the vial diameter $D = 25mm$ and by the intensity-to-concentration scaling factor $F_{I2C} = 214.6308 \frac{mm^3}{mmol \cdot mm}$. $F_{I2C}$ was calculated from the linear fit in *Figure 1c*. In each fitting iteration, simulated data was produced according to the diffusion-reaction-advection model above. Within the fitting procedure, $F_{I2C} \cdot (o(x, t) + f(x, t) + b(x, t))$ was fitted to $u_i^b(x, t)$ in a least square displacement sense within the domain $x \in [100, 700]$ μm from the sample surface and $t \in [0, 1200]$ min by using the *trust-region-reflective* algorithm of the nonlinear least-squares solver *lsqnonlin* in MATLAB to optimize the N=6 model parameters $D^e$, $k^{on}$, $k^{unmask}$, $S^0$, $M^0$ and $H$. These parameters were fitted to the average experimentally measured osmium accumulation from n=13 samples within the domains $x \in [100, 700]$ μm and $t \in [0, 1200]$ min. To evaluate the quality of the model fits, the residual standard error was calculated as $SE_{res} = \sqrt{\frac{\sum_{(x,t)} \left[ u_i^b(x,t) - (F_{I2C} \cdot (o(x,t) + f(x,t) + b(x,t))) \right]^2}{df}}$, were $df$ corresponds to the degree of freedom.

## Estimation of the average double-bond/ osmium binding site density in the mouse brain

The average phospholipid density in the mouse brain is 56,241±4000 nmol/g (*Barceló-Coblijn et al., 2007*) and for an average mouse brain volume of 508.91±23.42 mm³ (*Badea et al., 2007*) with a weight of 0.427±0.024 g (*The Jackson Laboratory, 2021*), a cubic millimeter brain tissue contains about 47.19 nmol phospholipids. Lipids in the brain carry on average 2.5 double bonds (*Fitzner et al., 2020*) which results in 117.975 nmol/mm³ double bonds to which osmium tetroxide could bind through osmium mono- or di-ester formation. Our proposed model predicts 2.999 times more binding sites (353.774±27.460 nmol/mm³), even without unmasking (*Figure 2—figure supplement 4*).

## Acknowledgements

We thank Michael Bozlar for advice on potassium ferrocyanide and osmium chemistry, Stefan N Oline for excellent assistance with animal perfusions and Alyssa M Wilson and Zhihao Zheng for comments on the manuscript. The authors acknowledge the use of Princeton's Imaging and Analysis Center, which is partially supported through the Princeton Center for Complex Materials (PCCM), a National Science Foundation (NSF)-MRSEC program (DMR-2011750). AAW acknowledges support by the CV Starr Fellowship in Neuroscience by the Princeton University. This work was supported by the

National Institutes of Health (NIH) grants U19 NS104648, NIH/NEI (1R01EY027036 to HSS), NIH/NINDS (1R01NS104926 to HSS), NINDS/NIH (U01NS090562 to HSS).

## Additional information

### Competing interests
David W Tank, H Sebastian Seung, Adrian Andreas Wanner: is an inventor of US Patent Application 16/681,028. The other authors declare that no competing interests exist.

### Funding

| Funder | Grant reference number | Author |
|---|---|---|
| CV Starr Fellowship in Neuroscience by the Princeton University | | Adrian Andreas Wanner |
| National Institutes of Health | NS104648 | David W Tank |
| National Institutes of Health | 1R01EY027036 | H Sebastian Seung |
| National Institutes of Health | U01NS090562 | H Sebastian Seung |

The funders had no role in study design, data collection and interpretation, or the decision to submit the work for publication.

### Author contributions
Sebastian Ströh, Eric W Hammerschmith, Conceptualization, Data curation, Validation, Investigation, Methodology, Writing – original draft, Writing – review and editing; David W Tank, H Sebastian Seung, Formal analysis, Supervision, Funding acquisition, Validation, Methodology, Writing – review and editing; Adrian Andreas Wanner, Conceptualization, Resources, Data curation, Software, Formal analysis, Supervision, Funding acquisition, Validation, Investigation, Visualization, Methodology, Writing – original draft, Project administration, Writing – review and editing

### Author ORCIDs
Sebastian Ströh  http://orcid.org/0000-0001-8128-2617
David W Tank  http://orcid.org/0000-0002-9423-4267
Adrian Andreas Wanner  http://orcid.org/0000-0002-5864-8577

### Ethics
Animal use procedures were approved by the Princeton University Institutional Animal Care and Use Committee (protocol number 2000) and carried out in accordance with National Institutes of Health standards (AAALAC International Institutional Number: Unit #1001, PHS assurance ID D16-00273).

### Decision letter and Author response
Decision letter https://doi.org/10.7554/eLife.72147.sa1
Author response https://doi.org/10.7554/eLife.72147.sa2

## Additional files

### Supplementary files
• Transparent reporting form

### Data availability
The code to analyze the X-ray projection images and to model the accumulation of heavy metals can be found on https://github.com/adwanner/XrayAssistedStaining, (copy archived at swh:1:rev:62c993

0d08bbdd1602614d515cd20b1cb1f89d15). All X-ray data is available for download on https://www.ebi.ac.uk/empiar/ with dataset ID EMPIAR-10782.

The following dataset was generated:

| Author(s) | Year | Dataset title | Dataset URL | Database and Identifier |
|-----------|------|---------------|-------------|-------------------------|
| Ströh S, Hammerschmith EW, Tank DW, Seung HS, Wanner AA | 2022 | In situ X-ray assisted electron microscopy staining for large biological samples | https://www.ebi.ac.uk/empiar/EMPIAR-10782/ | EMPIAR, EMPIAR-10782 |

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
