## [Editor Report]

This important study explores the kinetics of heavy metal staining of tissue using time-lapse imaging with X-ray micro computed tomography (CT). Introducing a compelling approach to investigate staining in situ, this work will be of interest to the wide community of scientists preparing biological samples in particular for large-volume electron microscopy. It will become a reference for the field in establishing a quantitative tool for assessing and developing staining protocols.

---

## [Decision Letter]

**Decision letter after peer review:**

Thank you for submitting your article "In situ X-ray assisted electron microscopy staining for large biological samples" for consideration by *eLife*. Your article has been reviewed by 3 peer reviewers, and the evaluation has been overseen by a Reviewing Editor and Catherine Dulac as the Senior Editor. The reviewers have opted to remain anonymous.

Essential revisions:

As you can see from the detailed reviews, all reviewers (and myself as well) felt that this work is potentially an important contribution.

However, as the central claim of the work is to use CT as a benchmark for the quality of staining for EM, it will be critical to directly relate those two measures, i.e. to perform EM imaging on previously CT imaged samples and relate the two measures of contrast to each other. This will be the critical experimental dataset for revision. There are other valuable comments below. Please edit the manuscript accordingly.

*Reviewer #1 (Recommendations for the authors):*

The authors are in a very good position to disambiguate current divergences in the initial steps of osmium staining of soft tissues for electron microscopy imaging purposes.

First, it would be very informative to assess the effect of a different iron salt commonly used to reduce osmium: potassium ferricyanide (e.g. by adding the experimental group 2% OsO4, 2.5% potassium ferricyanide to figure panels 1e-f).

Second, ref. Hua et al., 2015 implemented the osmium reduction posterior to the osmium incubation, by adding potassium ferrocyanide without intermediate washes. The authors have already explored this staining protocol in Supp Figure 5: samples being stained with OsO4 for 22h were transferred to a potassium ferrocyanide solution. It would be very informative to view those results in summary (ie averaging the 20h timepoint datasets across samples) along with the osmium and reduced osmium baselines shown in Figure 1e, and to complement the insight on sample expansion for that protocol, showing it against the same baselines, as in Figure 1f. An additional control group could be beneficial here: transferring the osmium-stained samples to a water vial, to account for decrease of staining by dilution and diffusion of unbound osmium. In this case it would not be expected to see sample degradation regardless of the incubation time.

Third, after staining with osmium or reduced osmium, samples are meant to be washed with water before proceeding to the next staining step, aiming to remove any unbound osmium from the media. The goal of optimizing the first staining step would therefore be to maximize the bound osmium remaining in the sample after the water wash(es). To that end, it would be interesting to see the evolution of intensity vs depth for samples placed in staining solution for 20h and then placed in water, for at least the most typical stains explored (osmium and reduced osmium).

Finally, it would be very useful if the authors released the code used for pre-processing single X-ray projections. Given the popularity of benchtop X-ray uCT devices, doing so could simplify and accelerate the adoption of this monitoring method of sample preparation protocols by the user community.

These additions could slightly improve the scope and completeness of an already robust and solid study.

*Reviewer #2 (Recommendations for the authors):*

Reference, page 4: Claude and Fullam (1945) used formaldehydes + osmium fixation. Only Sabatini introduced glutaraldehyde. It might be good to slightly modify the text.

Page 9: the authors refer to figure 1b to comment the staining after 20hours. Yet, fig 1b shows example at 10 hours max.

Page 9: "we found that for reduced osmium the accumulation of heavy metals peaks at a depth between 300- 800 μm (Figure 1e, Supplementary Figure 2), whereas the tissue above or below that depth is stained less." Could this be due to the anatomy/heterogeneity of the various cortical layers?

Page 9: about the shrinking/expansion of the tissue upon fixation with osmium or reduced osmium. This is very interesting as it could explain a few artifacts observed with several sample types. Yet in this experiment, the cortex biopsy is fitted inside a hole, of the same shape, made inside the sylgard support layer. It is thus possible that the tissue is constrained and can't expand in the lateral directions. Would this enhance the vertical expansion? And therefore such expansion would be less than 5%? Please comment on this.

In line with this, it might be good to adapt the drawings of figure 1a. The way the biopsy is displayed, it seems as if it was just placed onto the vial bottom, when in fact is is embedded in sylgard.

Page 12: about the osmium density: if I am not wrong these numbers are estimates, extrapolated from many assumptions re the density of unsaturated lipids, double bonds etc ... this should be made clear in the main text and not only in the methods. Even though the authors made their best possible efforts to describe the reactions within the tissues and to estimate the amount of binding sites, I doubt we have a solid proof of the actual concentration of bound or free osmium in this experiment.

Page 12: the authors write "While we assume that the tissue density is homogeneous across the cortical thickness ..." I don't think this is a fair assumption. The cortex is organized in layers of different cellular densities.

Page 14: discussion. what is not addressed in this paper is the secondary osmium blackening during dehydration. This has been described previously and attributed to unbound osmium and later reduction during the process. IT might be good to elaborate a bit on this. Of course, this leads to the comments made above, which highlight the need to check the samples in EM, i.e. to go, for some of them, through the full sample preparation.

Page 16: methods.

- The concentration ranges are surprising. Can you explain: "fixative solution 1.3 - 1.5% glutaraldehyde (GA) and 2.5 - 2.6% paraformaldehyde (PFA) in 0.14 - 0.15M cacodylate buffer with 2.0 - 2.1mM CaCl2"

- What is the rationale behind these incubation times. Intuitively, one would expect to keep whole brains longer in the fixative? "The samples were kept in fixative solution at 4 °C for 36 hours (4-mm biopsy punches) or for 12 hours (whole mouse brain)."

- About the sylgard molds: can you comment on the osmium infiltration. The sides of the biopsy might not be as well exposed as the top.

Page 17: X-ray microscopy: the working temperature is not indicate, though it is a very important parameter for the osmium fixation.

*Reviewer #3 (Recommendations for the authors):*

The manuscript would be strengthened if the authors compared the actual membrane contrast observed in cross-sections from their samples to demonstrate if these intensities can be quantitatively interpreted (beyond the description of staining gradients). I suggest to cut thin sections from a reduced osmium and osmium tissue cylinder and quantify the staining intensity/contrast across the 4mm diameter to compare with the x-ray profiles.

[Editors’ note: further revisions were suggested prior to acceptance, as described below.]

Thank you for resubmitting your work entitled "In situ X-ray assisted electron microscopy staining for large biological samples" for further consideration by *eLife*. Your revised article has been evaluated by Catherine Dulac (Senior Editor) and a Reviewing Editor.

The manuscript has been improved but there are some remaining issues that need to be addressed by small textual adjustments, as outlined below:

Please take note of the comments made by reviewer 3 regarding the strict correlation of X-ray and EM measure. Please adjust your textual description accordingly. While e.g. Mikula and Denk do provide some indication that this is the case (as does your work here), I would not want to give the impression that this is a question that has experimentally conclusively and quantitatively been settled. There is still room for direct experimental comparison for X-rays with EM under different conditions (although not necessarily in this manuscript).

In short, please tune down the corresponding comments and add a sentence or two in the discussion to mention the caveats and limitations discussed below and maybe point towards further such experiments. In this context it could be useful to expand a bit on the physics background and mention how X-ray contrast will depend e.g. on photon energy and that e.g. at the absorption edges the relationship of X-ray absorption and signal in the EM can become more complicated.

*Reviewer #2 (Recommendations for the authors):*

The revised version of the paper is efficiently addressing the points that I have raised and I am satisfied with the rebuttal answers as well.

In my opinion, the paper is ready for publication and expect the scientific community, especially the EM community to greatly benefit from this contribution.

*Reviewer #3 (Recommendations for the authors):*

My recommendation to the authors was to correlate their observed X-ray intensities to what is actually observed by EM in thin sections. In response, they cited Figure 4b from Mikula and Denk, 2015 as evidence that the X-ray reconstructed pixel intensities are correlated with what is observed in EM sections and state "Mikula and Denk (2015) showed that the pixel intensities of serial-section EM images and the intensity of the corresponding reslice are highly correlated."

I think this is an overstatement for several reasons:

1) As far as I can tell, Mikula and Denk Figure 4b is a n=1 measurement

2) The figure in question doesn't report a correlation coefficient, indeed their figure shows a peak SEM normalized intensity at the core of their sample and the X-ray normalized intensity has a peak at the edge of the sample

3) Mikula and Denk themselves are far more cautious in discussing this: "…X-ray microcomputed tomography, which exhibits an image contrast similar to that of SEM (Figure 4b). This allowed us to quickly test an embedded brain for defects and distortion before deciding whether to finish preparing it for electron microscopy." Mikula and Denk were clearing using X-ray reconstructions to check for gross defects/cracks, not to make quantitative comparisons.

Stroh et al., do provide a comparison of X-ray vs. EM intensities (new Figure 1 —figure supplement 1) with the caveat that the X-ray intensities are measured from a projection instead of a computed slice reconstruction. Perhaps due to this caveat the intensities do not appear to be very well correlated (although it would be preferable to actually show the correlation scatter plot and the measured correlation coeff). In light of this (and the above comments regarding the Mikula and Denk data), I remain unsure to what degree the measured X-ray intensities that form the foundation of the manuscript are informative about actual contrast in EM sections.

My original point was related to the author’' Figure 1e: the figure shows that reduced osmium stains tissue with a maximum intensity of ~1.1 (a.u.) compared to osmium alone at ~0.9 (a.u.). Because these intensities are not calibrated against the appearance of the staining in EM sections, their interpretation is limited. The authors state they were not able to test whether this particular finding is consistent with the intensity/contrast of EM sections due to embedding issues of reduced osmium samples.

---

## [Author Response]

Essential revisions:As you can see from the detailed reviews, all reviewers (and myself as well) felt that this work is potentially an important contribution.However, as the central claim of the work is to use CT as a benchmark for the quality of staining for EM, it will be critical to directly relate those two measures, i.e. to perform EM imaging on previously CT imaged samples and relate the two measures of contrast to each other. This will be the critical experimental dataset for revision. There are other valuable comments below. Please edit the manuscript accordingly.

We added Figure 1 —figure supplement 3 showing the ultrastructural preservation after 20 hours of incubation in OsO_4_ and Figure 1 —figure supplement 1 that shows the covariation of the EM and X-ray pixel intensities. Note, in the presented study we acquired X-ray projection images that represent the cumulative tissue and heavy metal density along the direction of projection through a 4 mm thick tissue punch. Therefore our X-ray projection pixel intensities cannot directly be compared to the EM pixel intensity of a thin section. As has been shown previously (see for example Figure 4b in Mikula and Denk 2015), in computed X-ray tomograms the intensity scales linearly with EM intensity, if the pixel intensity in an EM section is compared to the corresponding reslice of a computed X-ray tomogram.

Reviewer #1 (Recommendations for the authors):The authors are in a very good position to disambiguate current divergences in the initial steps of osmium staining of soft tissues for electron microscopy imaging purposes.First, it would be very informative to assess the effect of a different iron salt commonly used to reduce osmium: potassium ferricyanide (e.g. by adding the experimental group 2% OsO4, 2.5% potassium ferricyanide to figure panels 1e-f).

We added Figure 1 —figure supplement 7 and 8 as well as Figure 2 —figure supplement 3 that show the spatio-temporal accumulation of osmium reduced with 2.5% potassium ferricyanide K_3_[Fe(CN)_6_]. The accumulation of heavy metals for the buffered solution with 2% OsO_4_ and 2.5% potassium ferricyanide is not as homogeneous as for 2% OsO_4_ (Figure 1 —figure supplement 8a). But in contrast to the experiments where osmium was reduced with 2.5% potassium ferrocyanide, the sample does not shrink if incubated in buffered solution of 2% osmium reduced with 2.5% potassium ferricyanide (Figure 1 —figure supplement 8b). In contrast to potassium ferrocyanide, no ‘washout’ effect was observed if the sample was placed in buffered solution of 2.5% potassium ferricyanide after 20hrs of incubation in buffered 2% OsO_4_ (Figure 1 —figure supplement 8e).

Second, ref. Hua et al., 2015 implemented the osmium reduction posterior to the osmium incubation, by adding potassium ferrocyanide without intermediate washes. The authors have already explored this staining protocol in Supp Figure 5: samples being stained with OsO4 for 22h were transferred to a potassium ferrocyanide solution. It would be very informative to view those results in summary (ie averaging the 20h timepoint datasets across samples) along with the osmium and reduced osmium baselines shown in Figure 1e, and to complement the insight on sample expansion for that protocol, showing it against the same baselines, as in Figure 1f.

We added a corresponding Figure 1 —figure supplement 8.

An additional control group could be beneficial here: transferring the osmium-stained samples to a water vial, to account for decrease of staining by dilution and diffusion of unbound osmium. In this case it would not be expected to see sample degradation regardless of the incubation time.Third, after staining with osmium or reduced osmium, samples are meant to be washed with water before proceeding to the next staining step, aiming to remove any unbound osmium from the media. The goal of optimizing the first staining step would therefore be to maximize the bound osmium remaining in the sample after the water wash(es). To that end, it would be interesting to see the evolution of intensity vs depth for samples placed in staining solution for 20h and then placed in water, for at least the most typical stains explored (osmium and reduced osmium).

We added Figure 2 —figure supplement 4 and 5 that show the dynamics of heavy metal washout in double distilled H_2_O for 22 hours after 22 hrs of incubation in 2% buffered OsO_4_. First, the sample was quickly flushed with double distilled H_2_O four times to completely remove any buffered OsO_4_ solution. Subsequently, the sample was immersed in double distilled H_2_O for 22 hrs. The accumulation of heavy metals decreases by about 6% in a depth of 100-1200μm.

Our hypothesis is that this can be explained by unbound osmium that diffuses out of the sample and by the slight expansion of the sample (Figure 2 —figure supplement 5b). But in contrast to the washout effect in K_3_[Fe(CN)_6_], the reduction in heavy metal density appears to be small.

Finally, it would be very useful if the authors released the code used for pre-processing single X-ray projections. Given the popularity of benchtop X-ray uCT devices, doing so could simplify and accelerate the adoption of this monitoring method of sample preparation protocols by the user community.

The python and MATLAB code for processing the X-ray data is now available on https://github.com/adwanner/XrayAssistedStaining.

Reviewer #2 (Recommendations for the authors):Reference, page 4: Claude and Fullam (1945) used formaldehydes + osmium fixation. Only Sabatini introduced glutaraldehyde. It might be good to slightly modify the text.

We corrected the text and citations accordingly.

Page 9: the authors refer to figure 1b to comment the staining after 20hours. Yet, fig 1b shows example at 10 hours max.

We corrected it accordingly.

Page 9: "we found that for reduced osmium the accumulation of heavy metals peaks at a depth between 300- 800 μm (Figure 1e, Supplementary Figure 2), whereas the tissue above or below that depth is stained less." Could this be due to the anatomy/heterogeneity of the various cortical layers?

Indeed, the effects of reduced osmium accumulation could depend at least in parts of the tissue composition, i.e. the anatomy/heterogeneity of the cortical layers in the tissue block. It is, however, important to note that this effect is not present in non-reduced osmium immersed brain tissue samples that have the same composition. We added the sentences that ‘No such band is present in tissue immersed in non-reduced OsO_4_.’

Page 9: about the shrinking/expansion of the tissue upon fixation with osmium or reduced osmium. This is very interesting as it could explain a few artifacts observed with several sample types. Yet in this experiment, the cortex biopsy is fitted inside a hole, of the same shape, made inside the sylgard support layer. It is thus possible that the tissue is constrained and can't expand in the lateral directions. Would this enhance the vertical expansion? And therefore such expansion would be less than 5%? Please comment on this.In line with this, it might be good to adapt the drawings of figure 1a. The way the biopsy is displayed, it seems as if it was just placed onto the vial bottom, when in fact is is embedded in sylgard.

The contribution to the vertical expansion from the embedding in sylgard is expected to be minimal, because the sylgard layer was typically only 2 mm thick, whereas the height of the tissue samples was typically about 6.75 mm tall. The upper end of the sylgard layer can be seen in Figure 1b at the very bottom of the image (small arrowhead). Hence, only a small fraction of the sample sits in sylgard. We adapted Figure 1 accordingly.

That the contribution from the restriction in sylgard is small can also be seen from the following considerations:

2mm×π(a×124mm)2−2mm×π(124mm)2=2mm×(a−1)2×π(124mm)2

=2mm×(1−1/a)2×π(a×124mm)2

For buffered 2% OsO_4_ we measured an expansion of 5%, i.e. b=1.05 and according to 3. above the ‘true’ homogeneous expansion factor would have been a=1.049.

Page 12: about the osmium density: if I am not wrong these numbers are estimates, extrapolated from many assumptions re the density of unsaturated lipids, double bonds etc ... this should be made clear in the main text and not only in the methods. Even though the authors made their best possible efforts to describe the reactions within the tissues and to estimate the amount of binding sites, I doubt we have a solid proof of the actual concentration of bound or free osmium in this experiment.

The osmium density (as shown in Figure 2b) was measured using the pixel intensities which were calibrated using the concentration series shown in Figure 1c. No further assumptions are being made. In the discussion section highlighted by the reviewer we compare this measured osmium density to the density of double bonds of unsaturated lipids in the brain as estimated from numbers reported in the literature. The double bonds of unsaturated lipids are thought to be the main binding site for osmium. Comparing the measured density of osmium to the estimated density of binding sites we found that the measured osmium density is 2.999 times higher than the estimated density of double bonds in the brain. We therefore hypothesize that with long osmium incubation, 3 osmium atoms get associated with 1 double bond.

Page 12: the authors write "While we assume that the tissue density is homogeneous across the cortical thickness ..." I don't think this is a fair assumption. The cortex is organized in layers of different cellular densities.

We agree that the presented model and the underlying assumptions significantly simplify the actual bio-chemical environment. But despite these oversimplifications, the resulting model presented in Figure 2 captures the dynamics of osmium accumulation well. While modeling the tissue densities of different cortical layers in detail and calculating individual diffusion constants is beyond the presented study, we do agree that it would be very interesting to investigate how the diffusion constants differ in different tissue types. To illustrate this, we measured and modeled the staining kinetics of 2% buffered OsO_4_ in 4mm punches of liver tissue (Figure 2 - Figure Supplement 7). Interestingly, the effective diffusion coefficient appears to be >4X larger in liver tissue compared to brain tissue.

Page 14: discussion. what is not addressed in this paper is the secondary osmium blackening during dehydration. This has been described previously and attributed to unbound osmium and later reduction during the process. IT might be good to elaborate a bit on this. Of course, this leads to the comments made above, which highlight the need to check the samples in EM, i.e. to go, for some of them, through the full sample preparation.

We added Figure 1 - Figure Supplement 3that shows the ultrastructural preservation after 20 hours of osmium incubation. In addition, we performed additional experiments in which we measured how the osmium density changes during the H_2_O washing step after the OsO_4_ incubation. These results of the wash-out experiments are shown in Figure 2 - Figure Supplement 4 and 5. We do find a slight decrease in heavy metal density that can likely be explained on one hand by the wash out of unbound osmium from the sample as well as by expansion of the tissue which reduces the local heavy metal density.

Page 16: methods.- The concentration ranges are surprising. Can you explain: "fixative solution 1.3 - 1.5% glutaraldehyde (GA) and 2.5 - 2.6% paraformaldehyde (PFA) in 0.14 - 0.15M cacodylate buffer with 2.0 - 2.1mM CaCl2"

Some of the mice have been perfused by different experimenters that used slightly different effective fixative formulations.

- What is the rationale behind these incubation times. Intuitively, one would expect to keep whole brains longer in the fixative? "The samples were kept in fixative solution at 4 °C for 36 hours (4-mm biopsy punches) or for 12 hours (whole mouse brain)."

The 36 hours of post-perfusion incubation in fixative corresponds to the duration that we and other labs typically use for fixing brain tissue. The 12 hours post-perfusion incubation in fixative of the whole mouse brain might indeed be on the lower end for what might be necessary for thoroughly fixing a whole mouse brain after perfusion.

- About the sylgard molds: can you comment on the osmium infiltration. The sides of the biopsy might not be as well exposed as the top.

The sylgard layer was typically only 2 mm thick, whereas the height of the tissue samples was typically about 6.75 mm tall. The upper end of the sylgard layer can be seen in Figure 1b at the very bottom of the image (small arrowhead). Hence, only a small fraction of the sample sits in sylgard and the measurements have been taken several millimeters away from the sylgard, therefore we do not expect any major impact of the sylgard onto the osmium infiltration measurements. We adapted Figure 1 accordingly.

Page 17: X-ray microscopy: the working temperature is not indicate, though it is a very important parameter for the osmium fixation.

All staining and washing steps have indeed been done at room temperature. We added that information accordingly in the sample preparation section of the methods.

Reviewer #3 (Recommendations for the authors):The manuscript would be strengthened if the authors compared the actual membrane contrast observed in cross-sections from their samples to demonstrate if these intensities can be quantitatively interpreted (beyond the description of staining gradients). I suggest to cut thin sections from a reduced osmium and osmium tissue cylinder and quantify the staining intensity/contrast across the 4mm diameter to compare with the x-ray profiles.

We added Figure 1 —figure supplement 3showing the ultrastructural preservation after 20 hours of incubation in OsO_4_ and Figure 1 —figure supplement 1 that shows the covariation of the EM and X-ray pixel intensities. Note, in the presented study we acquired X-ray projection images that represent the cumulative tissue and heavy metal density along the direction of projection through a 4 mm thick tissue punch. Therefore our X-ray projection pixel intensities cannot directly be compared to the EM pixel intensity of a thin section. As has been shown previously (see for example Figure 4b in Mikula and Denk 2015), in computed X-ray tomograms the intensity scales linearly with EM intensity, if the pixel intensity in an EM section is compared to the corresponding reslice of a computed X-ray tomogram.

We also tried to acquire EM images of tissue punches stained with reduced osmium, but unfortunately the resin could not infiltrate the samples sufficiently well to allow us to collect serial sections.

[Editors’ note: further revisions were suggested prior to acceptance, as described below.]

The manuscript has been improved but there are some remaining issues that need to be addressed by small textual adjustments, as outlined below:Please take note of the comments made by reviewer 3 regarding the strict correlation of X-ray and EM measure. Please adjust your textual description accordingly. While e.g. Mikula and Denk do provide some indication that this is the case (as does your work here), I would not want to give the impression that this is a question that has experimentally conclusively and quantitatively been settled. There is still room for direct experimental comparison for X-rays with EM under different conditions (although not necessarily in this manuscript).In short, please tune down the corresponding comments and add a sentence or two in the discussion to mention the caveats and limitations discussed below and maybe point towards further such experiments. In this context it could be useful to expand a bit on the physics background and mention how X-ray contrast will depend e.g. on photon energy and that e.g. at the absorption edges the relationship of X-ray absorption and signal in the EM can become more complicated.Reviewer #3 (Recommendations for the authors):My recommendation to the authors was to correlate their observed X-ray intensities to what is actually observed by EM in thin sections. In response, they cited Figure 4b from Mikula and Denk, 2015 as evidence that the X-ray reconstructed pixel intensities are correlated with what is observed in EM sections and state "Mikula and Denk (2015) showed that the pixel intensities of serial-section EM images and the intensity of the corresponding reslice are highly correlated."I think this is an overstatement for several reasons:1) As far as I can tell, Mikula and Denk Figure 4b is a n=1 measurement2) The figure in question doesn't report a correlation coefficient, indeed their figure shows a peak SEM normalized intensity at the core of their sample and the X-ray normalized intensity has a peak at the edge of the sample3) Mikula and Denk themselves are far more cautious in discussing this: "…X-ray microcomputed tomography, which exhibits an image contrast similar to that of SEM (Figure 4b). This allowed us to quickly test an embedded brain for defects and distortion before deciding whether to finish preparing it for electron microscopy." Mikula and Denk were clearing using X-ray reconstructions to check for gross defects/cracks, not to make quantitative comparisons.Stroh et al., do provide a comparison of X-ray vs. EM intensities (new Figure 1 —figure supplement 1) with the caveat that the X-ray intensities are measured from a projection instead of a computed slice reconstruction. Perhaps due to this caveat the intensities do not appear to be very well correlated (although it would be preferable to actually show the correlation scatter plot and the measured correlation coeff). In light of this (and the above comments regarding the Mikula and Denk data), I remain unsure to what degree the measured X-ray intensities that form the foundation of the manuscript are informative about actual contrast in EM sections.My original point was related to the authors' Figure 1e: the figure shows that reduced osmium stains tissue with a maximum intensity of ~1.1 (a.u.) compared to osmium alone at ~0.9 (a.u.). Because these intensities are not calibrated against the appearance of the staining in EM sections, their interpretation is limited. The authors state they were not able to test whether this particular finding is consistent with the intensity/contrast of EM sections due to embedding issues of reduced osmium samples.

Thank you very much for the constructive feedback on the revised manuscript. We agree with Reviewer 3 that adding some quantitative evidence for the correlation between electron microscopy images and X-ray images would strengthen the revised manuscript. On one hand, we adjusted the wording regarding the similarity/correlation between EM and micro-CT in the introduction. On the other hand, we added a note of

caution to the discussion that for a qualitative comparison between EM and micro-CT appearance the two imaging modalities need to be calibrated and the precise image acquisition parameters need to be considered.

In addition, we added additional data comparing EM and micro-CT appearance similar to Mikula et al., to Figure 1 —figure supplement 1. In two different *en bloc* stained brain samples we show the correspondence between the EM and micro-CT appearance. As suggested by Reviewer 3 we included the (indeed high) correlation statistics and corresponding scatter plots.

To conclude, we have tuned down the claims regarding the correspondence between EM and X-ray imaging and at the same time added more examples and quantifications supporting the claims. We believe that these changes and additional information have improved the revised manuscript.